# Development and Evaluation of *Virola oleifera* Formulation for Cutaneous Wound Healing

**DOI:** 10.3390/antiox11091647

**Published:** 2022-08-25

**Authors:** Glaucimeire R. Carvalho, Débora S. Braz, Talita C. O. Gonçalves, Rafaela Aires, Larissa Z. Côco, Marcio Guidoni, Marcio Fronza, Denise C. Endringer, Antonio D. S. Júnior, Manuel Campos-Toimil, Breno V. Nogueira, Elisardo C. Vasquez, Bianca P. Campagnaro, Thiago M. C. Pereira

**Affiliations:** 1Pharmaceutical Sciences Graduate Program, Vila Velha University (UVV), Vila Velha 29102-920, Brazil; 2Physiology and Pharmacology of Chronic Diseases (FIFAEC), Center for Research in Molecular Medicine and Chronic Diseases (CiMUS), University of Santiago de Compostela, 15782 Santiago de Compostela, Spain; 3Laboratory of Translational Physiology, Physiological Sciences Graduate Program, Federal University of Espírito Santo, Vitoria 29047-105, Brazil; 4Federal Institute of Education, Science and Technology (IFES), Vila Velha 29106-010, Brazil

**Keywords:** healing, plant extract, polyphenols, oxidative stress, antioxidant, bicuíba, reepithelization

## Abstract

In regions adjacent to the Brazilian Atlantic Forest, *Virola oleifera* (VO) resin extract has been popularly used for decades as a skin and mucosal healing agent. However, this antioxidant-rich resin has not yet been investigated in wound healing, whose physiological process might also be aggravated by oxidative stress-related diseases (e.g., hypertension/diabetes). Our aim, therefore, was to investigate whether VO resin presents healing effects through an innovative cream for topical applications. For this, adult male Wistar rats were divided into four groups. Then, four 15 mm excisions were performed on the shaved skin. All treatments were applied topically to the wound area daily. At the end of experiments (0, 3rd, and 10th days) macroscopic analysis of wound tissue contraction and histological analysis of inflammatory cell parameters were performed. The group treated with VO cream showed the best wound contraction (15%, *p* < 0.05) and reduced levels of lipid peroxidation and protein oxidation (118% and 110%, *p* < 0.05, respectively) compared to the control group. Our results demonstrated the healing capacity of a new formulation prepared with VO, which could be, at least in part, justified by antioxidant mechanisms that contribute to re-epithelialization, becoming a promising dermo-cosmetic for the treatment of wound healing.

## 1. Introduction

*Virola oleifera* (VO) belongs to the Myristicaceae family, an abundant tree in the Brazilian Atlantic Forest [1,2]. They are popularly known as ‘bicuíba’, ‘bicuíva’, ‘candeia-do-caboclo’, ‘sangue de boi’, or ‘ucuúba’ (word of Tupi origin “uku”= tallow, grease, fat and “uba”= plant, tree), meaning that trees produce a resin substance [3,4]. In folk medicine, this red resin obtained from the trunk of the VO is used to stop bleeding and is also used in wound healing and arthritis, as well as inflammatory conditions [1,4]. More recently, this resin from VO (RV) has demonstrated a relative abundance of antioxidant molecules (phenolic acids, tannins, and flavonoids) [1,3] and different pharmacological activities, including antioxidant, gastroprotective [1], atheroprotective [5], nephroprotective, and anti-apoptotic properties [3]. However, its healing properties in skin wounds have not yet been evaluated in experimental models.

Wound healing is a physiological reaction to tissue injury [6,7], characterized by four continuous and overlapping phases: hemostasis, inflammation, proliferation, and remodeling [8,9]. However, it is known that imbalances in one or more of these phases can compromise the healing process. For example, persistent inflammatory and oxidative stress (common hallmarks of most chronic degenerative diseases such as diabetes and/or hypertension) generate vascular and immunological complications, slowing down wound closure and healing [10,11].

Specifically, it is known that wounds are heterogeneous, and the healing process is multifactorial, being influenced by many internal and external factors. Therefore, several in vitro and in vivo assays are available to evaluate wound healing [6,7]. First, in vitro parameters include the evaluation of different cell types (e.g., fibroblasts, macrophages, keratinocytes) and the observation of cellular activity such as proliferation, migration, inhibition of inflammatory mediators, and extracellular matrix formation. Although the in vitro tests have yielded promising results as valuable and inexpensive tools to find new species with healing effects, the in vivo assays (e.g., macroscopic observations, biochemical and histological studies of the healing) are key as a proof of concept [8,12].

In the last few decades, several plant species rich in phenolic compounds as well as flavonoids have demonstrated wound healing properties. As an interesting example, we can mention the *Struthanthus vulgaris* (also used in traditional folk medicine in Brazil), demonstrating antioxidant, anti-inflammatory, and re-epithelialization activities [13,14]. In addition to unique species, vegetable oil blend formulations containing sunflower oil (*Helianthus annuus*), olive oil (*Olea europaea*), rosehip oil (*Rosa aff. rubiginosa*), linseed oil (*Linum usitatissimum*), black currant oil (*Ribes nigrum*), and macadamia oil (*Macadamia ternifolia nut oil*) also decrease the release of proinflammatory cytokines, reduce the migration of polymorphonuclear cells to the wound site, and promote proper deposition of the extracellular matrix, revealing an interesting healing effect [15]. In this study, the objective was to investigate whether a cream containing VO resin has healing activity in skin wounds in rats, justified at least in part, due to the high antioxidant effect.

## 2. Materials and Methods

### 2.1. Resin Material

The resin from VO was collected in November 2019 (spring season) from the district of Fazenda Guandu (20°13490′ S, 041°06692′ W, Afonso Claudio, Espirito Santo, Brazil) with authorization (IEMA 629/09) and in accordance with Brazilian law (Resolution 29, 12 June 2007). After making 0.5 cm deep incisions in the tree trunk, 20 mL of the fluid was collected in an amber glass bottle and kept at 4 °C until the final analysis (Figure 1A). The fluid exudate was dried at 45 °C for 72 h, then granulated (Figure 1B), and the final yield obtained was 42 g of dry resin [1,3,5].

### 2.2. Animals

All experiments involving the use of Wistar rats were conducted in agreement with the Brazilian Animal Care Committee and were approved by the Committee of Ethics, Bioethics, and Animal Welfare of the Vila Velha University (UVV) (CEUA-UVV protocol 527/2019). A total of 60 male adult animals (300–370 g, 6–8 weeks old) were housed under controlled conditions with a 12 h/12 h light/dark cycle under standard conditions of temperature and humidity. The rats were fed a normal chow diet and water ad libitum and kept in individually cages (to prevent biting and fighting).

### 2.3. Cream Composition

The creams formulated specifically for this study are shown in Figure 2. Firstly, the cream VO 5% was incorporated with 10% Compritol ATO 888 (a non-irritant lipid excipient composed of a blend of different esters of behenic acid with glycerol). A prototype formulation containing RV was subjected to preliminary pharmaceutical development using different concentrations (containing 1–5% *w*/*w*), being defined as the concentration of 5%. In the pilot study, concentrations above 5% did not allow good cream stability, probably due to excess amphoteric substances in the resin. This second incorporation of VO 5% was in a vegetable oil blend (Vegederm), also developed by Guidoni et al. [15], which previously demonstrated a healing effect in skin-incised rats. The positive control used was the commercially available Dersani^®^ (linoleic, capric, caproic and caprylic acids, sunflower oil, soy lecithin, vitamin A and E). Mineral oil was formulated with the same gelling agent used in the VO formulation to obtain the same ointment consistency to be used as the vehicle control.

### 2.4. Excisional Wound Model

After 1 week of acclimatization, the animals were anesthetized with ketamine (Ceva, São Paulo, Brazil, 80 mg/kg i.p.) and xylazine (Rompum, Bayer, São Paulo, Brazil, 10 mg/kg i.p.), as used by others [16,17]. To create wounds, the skin was previously cleaned with 70% ethanol. Using a 15-mm punch biopsy instrument (Miltex, Inc., York, PA, USA), 4 full-thickness skin wounds were created on the dorsum region of each animal. Immediately after surgery, all animals received analgesic metamizole (100 mg/kg, in water) for pain relief for three days.

### 2.5. Experimental Group

The animals were randomly distributed into four groups (*n* = 5 each). The treatments were: vehicle control (VC) treated with vehicle formulation, positive control (PC) treated with Dersani^®^, *Virola oleifera* (VO) treated with cream VO 5%, and *Virola oleifera* + Vegederm (VO + Veg) treated with VO 5% incorporated into oil blend. Immediately after injury, all animals were submitted to the respective four different types of treatments. As after surgery the borders developed variability until day four, we decided to initiate the evaluation only on the fourth day, considering in this study the day “zero” (avoiding possible bias in our investigation). Subsequently, the 2nd and 4th, 6th, 8th, and 10th were also evaluated.

### 2.6. Wound Size Measurement and Score Wound

The wound areas were measured using ImageJ software (NIH, USA). Wound closure represents the percentage of wound reduction from wound size, following the formula:((wound area day 0 − wound area at days 0, 2, 4, 6, 8, and 10)/wound area day 0) × 100
according to Caetano et al. [18], and Jamadagni et al. [19]. The values were expressed as the percentage of wound healing. The percentage of contraction by treatment group was analyzed based on the calculation of the area under the curve (AUC) using the Prism software (Prism 6.0, GraphPad Software, Inc., San Diego, CA, USA) and was expressed in arbitrary units. The scoring method for each wound was given in semi-quantitative evaluation of bleeding, borders, and crust, performed according to the study of Jamadagni et al. [19], adapting the evaluation to the three stages: 0, 4th, and 10th day.

### 2.7. Biochemical Analyses

After the 10th day, all animals were euthanized with thiopental (Cristalia, São Paulo, Brazil, 100 mg/kg, i.p.) for collected blood and skin samples. The plasma was obtained by centrifugation at 480 g for 10 min from blood and kept at −80 °C for oxidative stress and biochemical analysis. Serum concentrations of ALT, AST, urea, urea nitrogen, and serum creatinine were measured using an automatic spectrophotometer (AU 400/680, Olympus/Beckman Coulter, Munich, Germany) from a clinical analysis laboratory (Tommasi Laboratory, Vitoria, Brazil).

### 2.8. Histological Analyses

Just as was done with score wound and biochemical analyses, two tissues from each rat in each group of treatment were collected (0, 4th, and 10th day) to evaluate the histopathological typical alterations. The tissue samples were fixed in 3.7% paraformaldehyde (in 0.1 mol/L PBS, pH 7.4), and mounted into paraffin. They were then serially sectioned at 6 μm and stained with hematoxylin–eosin (H&E). Finally, images were taken with software (Honestec VHS to DVD 3.0 SE) fashion at 100× using a digital camera attached to a light microscope (Model Leica Microscope Type 501095).

### 2.9. Thiobarbituric Acid Reactive Substances (TBARS)

The concentration of thiobarbituric acid (TBA) reactive substances in plasma and skin tissue homogenate (1:10) was determined according to Coutinho et al. [5]. The method involves the reaction between malondialdehyde (MDA) and TBA, resulting in products of lipid peroxidation at a temperature of 95 °C and read at 532 nm using a spectrophotometer (Spectra-MAX-190, Molecular Devices, Sunnyvale, CA, USA). The protein parameter was quantified as µmol of MDA/mg of protein, using the same Bradford method [20].

### 2.10. Advanced Oxidation Protein Products (AOPP)

AOPP was prepared in tissue following the method described by Hanasand et al. [21] using spectrophotometry by the formation of triiodide ion through the oxidation of KI with chloramine-T. A total of 200 μL of homogenate of skin (200 mg diluted 1:5 in PBS, *w*/*v*) was combined with KI (1.16 mol/L, 10 µL) and acetic acid (0.20 mol/L, 160 µL) and placed in the well of a 96-well microtiter plate (Becton Dickinson Labware, Lincoln Park, NJ, USA). The sample was mixed on a plate shaker and the absorbance of the reaction was read at 340 nm in a microplate reader (Spectra-MAX-190, Molecular Devices, Sunnyvale, CA, USA). The concentrations were expressed in μmol/mg of total protein, previously quantified by the Bradford method [20].

### 2.11. In Vitro Assays: ROS Determination

#### 2.11.1. Superoxide Anion Quantification

Superoxide anion was measured by the NBT reduction assay, as previously described [22,23]. Briefly, monocyte/macrophage-like cells (RAW 264.7) were seeded at 7.0 × 10^4^ cells/mL in 96-well tissue culture plates (overnight), and the treatments of the cells were realized using different concentrations (3.13, 6.25, 12.5, 25, 50, and 100 µg/mL) of RV for 1 h. Subsequently, the cells were stimulated with LPS (1 µg/mL) for an additional 20 h. Tempol (12 mM) was used as a positive control. After incubation, 100 μL of a NBT solution (1 mg/mL) was added and incubated (5% CO_2_ at 37 °C) for 1 h. The incubation medium was then removed and the cells were washed and lysed with KOH (2 mol/L):DMSO (1:1). The absorbance of reduced NBT was measured at 630 nm in a microplate reader (Molecular Devices Spectra, Sunnyvale, CA, USA). The experiments were performed in triplicate (*n* = 3) and the results were expressed as a percentage of the control without LPS.

#### 2.11.2. NO Production

Nitric oxide (NO) concentration was quantified as its biotransformed product of nitrite in LPS stimulated RAW 264.7 supernatant cells according to the Griess reaction [24,25] under simple modifications [5,23]. As well as the previous assay, RAW 264.7 cells were also plated at 7.0 × 10^4^ cells/mL in 96-well tissue culture plates (overnight) and later treated without or with similar crescent concentrations (3.13, 6.25, 12.5, 25, 50, and 100 µg/mL) of RV for 1 h. Subsequently, the cells were stimulated with LPS (1 µg/mL) for an additional 20 h, then L-NIL (L-N6-(1-iminoethyl)lysine, a selective inhibitor of inducible nitric oxide synthase) was used as a positive control (50 μM). The cell supernatant was then used for the quantification of nitrite using Griess reagent (1% sulfanilamide in 5% H_3_PO_4_ and 0.1% N-(1-naphthyl)ethylenediamine in distilled water). The absorbance was measured at 540 nm in an ELISA plate reader (Spectra-MAX-190, Molecular Devices, Sunnyvale, CA, USA) and the inhibitory rates were calculated using a standard calibration curve prepared with sodium nitrite (Sigma-Aldrich, St. Louis, MO, USA). The results were obtained by nitrite concentration (μM) calculated by regression analysis using a standard curve of sodium nitrite (0–200 μM). Absorbance at 540 nm was determined using a microplate reader (Molecular Devices Spectra, USA). The experiments were performed in triplicate (*n* = 3).

### 2.12. Statistical Analysis

All results were expressed as mean ± SEM (Standard Error of Mean). Data analysis was performed by one-way analysis of variance (ANOVA), followed by Tukey’s post hoc test, using Prism software (Prism 6, GraphPad Software, Inc., San Diego, CA, USA). To compare data on the wound area and percent contraction, we used two-way ANOVA. Categorical variables are presented as frequencies and were compared using Fisher’s exact test. The differences were considered significant when *p* < 0.05.

## 3. Results

### 3.1. Biochemical and Body Weight of Experimental Groups

As can be seen in the tables below, neither body weight (Table 1) nor serum biochemical parameters (Table 2) showed significant differences between groups (*p* > 0.05) during 10 days of treatment. Therefore, the induction of skin lesions or their treatments did not reflect on systemic effects such as ponderal parameters, liver integrity, or renal function.

### 3.2. Healing Effect of Virola Cream

The typical images of the wounds show a healing effect for the VO group compared to other experimental groups (Figure 1A). Through quantitative analysis (Figure 1B), the VO group reduced wound area (76%) from the third day (VC: 88% and PC: 94%, *p* < 0.05) until the end of the treatment. On the 8th day, the wound contraction in the VO group was higher (~50%) compared to all other treatments, and after 10 days was observed on total wound closure only in the VO group (Figure 3A, last column). In Figure 3C, the area under the curve graph reveals a greater efficiency of VO in the healing when compared to the VC and PC groups (27% and 23%, respectively, *p* < 0.05).

### 3.3. Dryness Score

Figure 4A shows typical photographs of the dryness of wounds in each group. As can be seen, at 0 and 4 days, all treatments (PC, VO, or VO + Veg) were able to reduce early bleeding compared to the VC group (Figure 4B). Interestingly, the VO groups showed, from day 0, better bleeding control compared to the other groups.

### 3.4. Borders Score

Figure 5A displays the progression of the wound borders in the experimental groups. On day 0, the VC and PC groups showed visible edema in the wound borders. On the other hand, the VO and VO + Veg groups exhibited wound borders without irregularities. On day 4, the maintenance of the borders without abnormalities was observed only in the VO group until the end of treatment (*p* < 0.05, Figure 5B). Although VO alone has demonstrated a good healing effect, the VO + Veg group also showed a better border score compared to the VC group (*p* < 0.05, Figure 3B).

### 3.5. Crust Score

Figure 6 shows that the treatment with VO was higher on days 0 and 4, with satisfactory crust formation when compared to the other experimental groups (*p* < 0.05). In parallel, the VO + Veg group exhibited better crust formation compared to the VC and PC groups (*p* < 0.05). However, at the 10th day of treatment, only the VO group showed a good formation of secondary crust, being significantly higher than the VC and VO + Veg groups (*p* < 0.05).

### 3.6. Histology

Figure 7 demonstrates the typical histological analysis using the tissue biopsy specimens of each group throughout the treatment. On the first day of treatment (day 0), the VO group showed reduced granulation tissue when compared to the other groups. After 4 days, only VC and PC groups demonstrated the presence of perivascular inflammatory infiltrate in the dermis and greater collagen disorganization (red arrows). At the end of the treatment, the groups VC and PC showed less re-epithelialization and keratinization when compared to VO.

### 3.7. Lipid Peroxidation (TBARS) in Skin

Thiobarbituric acid-reactive substance (TBARS) measurements were taken on 0 and 10 days after the treatment to obtain the difference (delta) between the last and the first day of treatment (Figure 8). The PC group (PC: −124.3 ± 70, µmol MDA/mg protein, *p* < 0.05) and treatment with VO (−84 ± 22 µmol MDA/mg protein, *p* < 0.05) were able to reduce lipid peroxidation compared to the VC group (443 ± 226 µmol MDA/mg protein). However, the VO + Veg group showed no difference (VO + Veg: 162 ± 48 µmol MDA/mg protein).

### 3.8. Advanced Oxidation Protein Products (AOPP) in Skin

The tissue concentrations of AOPP in the experimental groups were measured at 0 and 10 days after the treatment. The results for protein oxidation (Figure 9) are similar to that observed for lipid peroxidation. The PC group (PC: −1970 ± 1160, µmol/mg of protein, *p* < 0.05), and the VO (−438 ± 108 µmol /mg of protein, *p* < 0.05) were able to reduce protein oxidation compared to the VC group (4267 ± 1393 µmol/mg protein). However, the VO + Veg group also showed no difference (1273 ± 577, µmol/mg of protein).

### 3.9. In Vitro Assays: ROS Determination

#### 3.9.1. Superoxide Anion Quantification

Figure 10 demonstrates the capability of RV to inhibit superoxide anion production induced by LPS in monocyte/macrophage cells. Firstly, as expected, the superoxide production in these cells was increased due to LPS exposition (~6-fold) and was prevented by Tempol (~3-fold, *p* < 0.05). Concerning RV, only doses over 50 µg/mL resulted in a diminution of superoxide anion availability compared to the LPS induced group (V50: ~35%; and V100: ~50%, *p* < 0.05).

#### 3.9.2. NO Production

Figure 11 demonstrates the capability of RV to inhibit NO production induced by LPS also in monocyte/macrophages cells. The nitrite accumulation in these cells was also increased due to LPS exposition (~8-fold) and was decreased by NO synthase (L-NIL) (~1-fold, *p* < 0.05). Interestingly, cells simultaneously exposed to treatment with LPS and crescent concentrations of RV resulted in a diminution of NO availability in a concentration-dependent manner, with a significant reduction (compared to the LPS group) from 12.5 µg/mL (V12.5: ~1.6-fold; V25: ~1.8-fold; V50: ~2.3 fold; and V100: ~4-fold, *p* < 0.05).

## 4. Discussion

VO has been used in folk medicine for different oxidative stress-related disorders such as stopping bleeding and wound healing and has also been applied to inflammatory conditions, mainly in middle-aged metabolic syndrome patients. Recent investigations from our research group have reported that VO resin has several biological activities, such as anti-apoptotic, anti-inflammatory, nephroprotective, gastroprotective, and anti-atherosclerotic effects [1,3,5]. In this study, we observed that animals treated topically with an innovative cream with 5% VO for 10 days demonstrated a healing effect accompanied by antioxidant activity, justified by both tissue antioxidant effects (skin) and in vitro tests using macrophages.

It is well established that excessive ROS bioavailability has been associated with impaired wound repair in non-healing or chronic wounds [25,26,27,28]. More specifically, effects at the molecular level reveal that ROS can upregulate pro-inflammatory cytokine and metalloproteases, modifying extracellular matrix proteins, impairing dermal fibroblast, and compromising keratinocyte function [28,29,30]. In parallel, recent reviews show that polyphenols in chronic wounds are capable of reducing ROS availability by activating pro-healing and anti-inflammatory gene pathways [31]. Therefore, new therapies based on topical extracts rich in antioxidants might be interesting adjuvant strategies to regulate the redox balance, allowing the wound to continue the natural phases of healing, which justifies the relevance of our preclinical study.

Previous chemical investigations from our research group using Liquid chromatography-mass spectrometry (LC/MS) and Electrospray ionization (ESI), coupled with Fourier-transform ion cyclotron resonance mass spectrometry (FT-ICR MS) have demonstrated that VO resin is composed of a mixture of antioxidant compounds, such as polyphenols and flavonoids (and more precisely ferulic and gallic acid) [1,3,5] that promote its protective effects through antioxidant pathways. Firstly, Bôa et al. [3] showed that VO resin (administered by the oral route) was able to reduce the levels of ROS promoting nephroprotection by tissular antioxidant mechanisms. Secondly, Pereira et al. in 2017 [1] also revealed the important antioxidant activity of the same VO through in vitro analysis (FRAP, DPPH, ABTS), suggesting that the gastroprotective effect from VO observed in mice (also via oral gavage) could occur by its antioxidant effects. Given the topic of the healing effects observed in this study by VO, the strategy of verifying the role of antioxidant effect on the skin is important and innovative.

In the present research, we demonstrated the antioxidant potential of VO, as shown by indirect biomarkers of lipid peroxidation and oxidized proteins on the skin (TBARS and AOPP, respectively) and in macrophages. Just as in our study, other investigations that used different plant extracts with antioxidant activity have also demonstrated benefic effects treating wounds in vivo [32,33,34], reinforcing this link between oxidative stress and healing. Our data therefore highlight the relevance of ROS bioavailability in tissues, allowing the expansion of new screenings for healing in experimental models.

The aim of the insertion of the VO and Vegederm group was to demonstrate whether there would be any synergistic healing effect with the incorporation of VO in a dermo-cosmetic base (rich in antioxidants) with healing action previously developed by our research group [15]. Through the results obtained, this possible additional benefit was not observed in other studies [35,36,37]. It is well known that excessive exposure to antioxidants could also culminate in a pro-oxidant effect, impairing the healing process, as found in our study. Therefore, our study expands on the idea that the introduction of several antioxidant products in a formulation may not always trigger additional effects, being fundamental to investigating these new formulations in experimental models before any large-scale production efforts.

Regarding tissue contraction, it is known that a series of important events must happen previously during the proliferative phase, such as angiogenesis, formation of granulation tissue, and re-epithelialization that results in tissue contraction [38,39,40]. In our study, the histological data demonstrate that the topical application of VO presents a lower amount of granulation tissue, which is probably due to the healing acceleration process 3 days before the beginning of the evaluation. Another consequence was to observe the good evolution of those with the organization of collagen fibers, demonstrating greater re-epithelialization and keratinization at the end of the treatment evaluation. Given this data, we suggest that VO may influence the migration of fibroblasts to the lesion site, thus contributing to re-epithelialization. This hypothesis can also be supported by previous studies from our laboratory to demonstrate that VO does not have cytotoxic effects on fibroblasts, as demonstrated by in vitro analysis [5]. In addition, our data corroborate other studies that demonstrate the ability of polyphenols to promote re-epithelialization [41,42] Thus, in our study the histological data corroborate the macroscopic results, demonstrating tissue contraction in vivo through the topical use of VO, as well as its antioxidant action (in vivo and in vitro). Future investigations on collagenization and cell viability are needed to elucidate other possible activities of VO in the healing process.

Our study opens up opportunities for future research in the translational area to validate the clinical investigations of VO as a potential healing agent. Moreover, these data highlight the importance of ethnopharmacology for the ‘discovery’ of new bioactive products in Brazilian soil, encouraging the preservation of VO species, and, at the same time, stimulating local/regional economic activity.

Although these results are promising, it is important to consider that our results were obtained with the gross (total) part of the resin. We suggest that future investigations with different VO extracts (hexane, chloroformic, ethanolic, aqueous) should be performed to identify potential active substances with healing action contained in the VO. However, we cannot rule out the hypothesis that the healing effect observed only occurs due to the synergism of the “pool” of active substances contained in the VO, thus justifying the applicability only of the crude extract (but not fractions) of the same species [43,44,45].

## 5. Conclusions

Under in vivo and in vitro experimental conditions, our study demonstrates the healing capacity of an innovative formulation prepared with VO. Through the biochemical analyzes performed, we suggest that VO acts, at least in part, by antioxidant mechanisms, contributing to the reduction of oxidative stress and re-epithelialization. Thus, VO resin can be a promising natural therapeutic agent for the treatment of wound healing, in both veterinary and human fields.

## Figures and Tables

**Figure 1 antioxidants-11-01647-f001:**
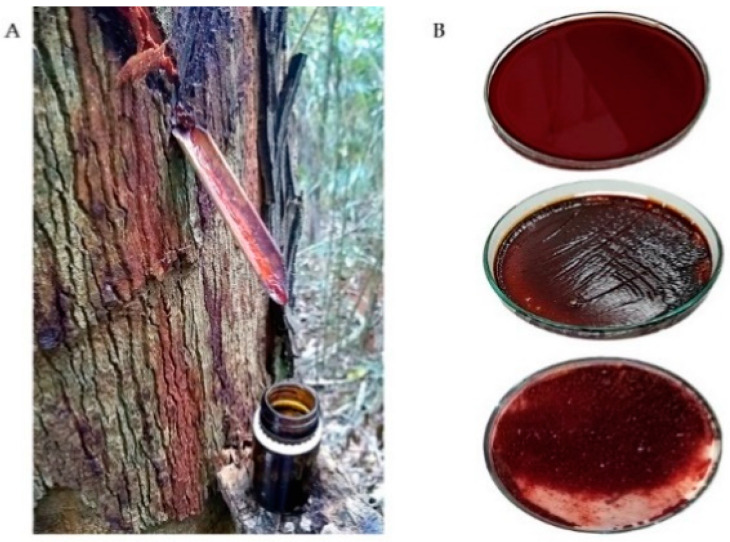
Extraction of *Virola oleifera* resin. After 0.5 cm deep incisions were made in the tree trunk, the fluid was collected in an amber glass bottle, as showed in (**A**). The fluid exudate was dried (45 °C for 72), then granulated, until obtaining the dry resin, as seen in (**B**).

**Figure 2 antioxidants-11-01647-f002:**
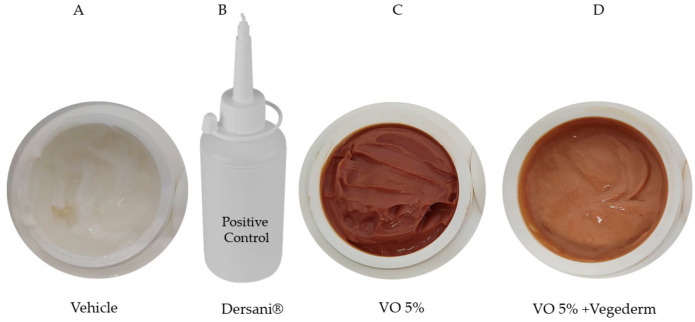
Four types of creams were formulated in this study. Vehicle (**A**), Dersani^®^ (**B**), VO 5% (based on the pilot study) (**C**), and the second incorporation of VO 5% into vegetable oil blend (Vegederm) (**D**).

**Figure 3 antioxidants-11-01647-f003:**
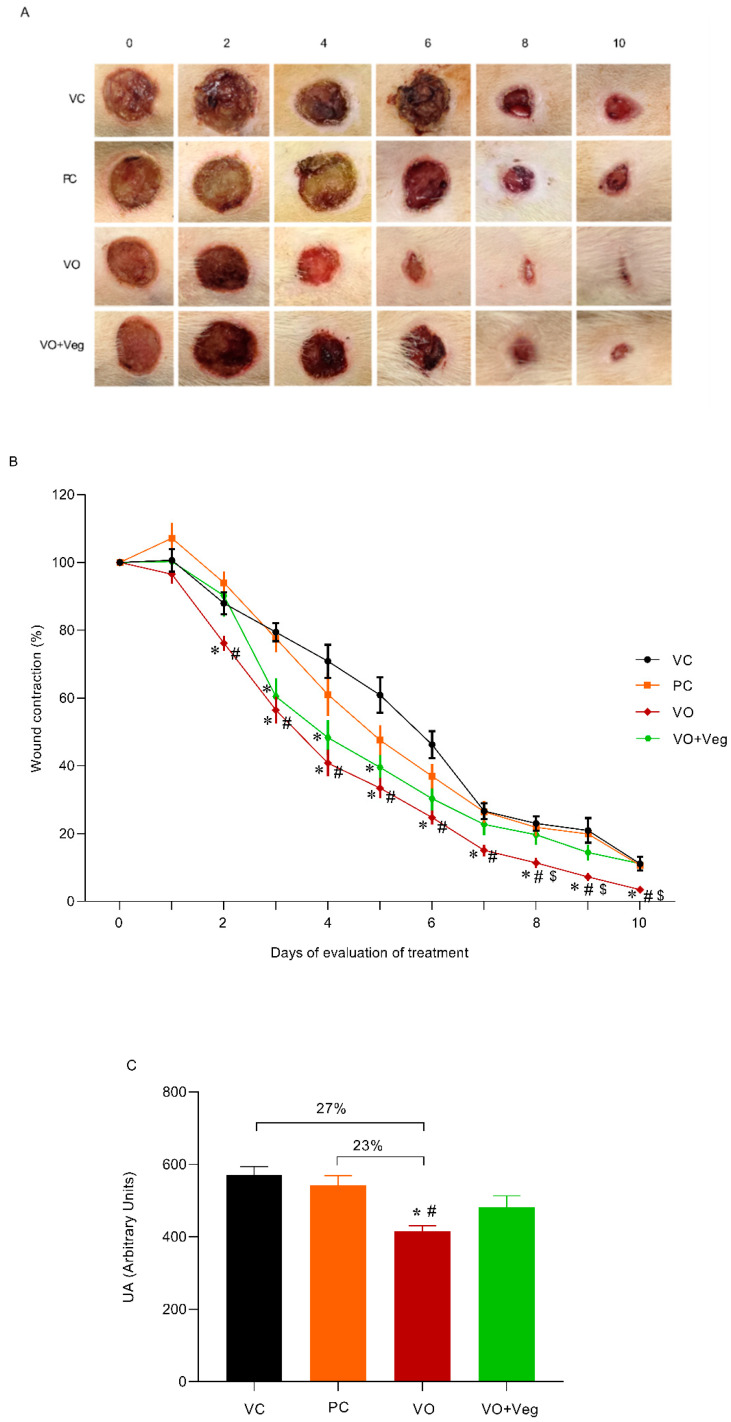
(**A**) Representative photographs of the wound by group and treatment evaluation time (0 until 10th day). (**B**) Quantitative analysis of wound contraction after daily topical use of vehicle control (VC), positive control Dersani^®^ (PC), *Virola oleifera* (VO), and *Virola oleifera* with Vegederm (VO + Veg) for 10 days. (**C**) Area under the curve of the experimental groups representing healing effect between groups. The data are expressed as mean ± SEM; * *p* < 0.05: VO vs. VC; # *p* < 0.05: VO vs. PC, and $ *p* < 0.05: VO vs. VO + Veg.

**Figure 4 antioxidants-11-01647-f004:**
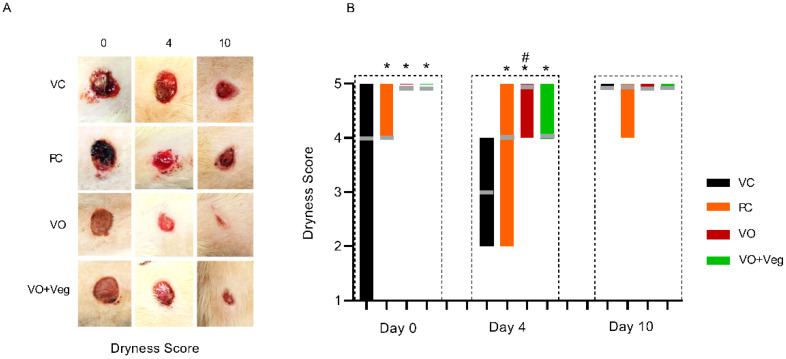
Typical images of dryness in respective groups and treatment evaluation time (**A**). Graph of dryness scores in VC, PC, VO, and VO + Veg groups, in three periods of evaluation. The bars represent the data dispersion (minimum and maximum), and the horizontal gray lines represent mode (**B**). * *p* < 0.05: VO vs. VC; # *p* < 0.05: VO vs. PC.

**Figure 5 antioxidants-11-01647-f005:**
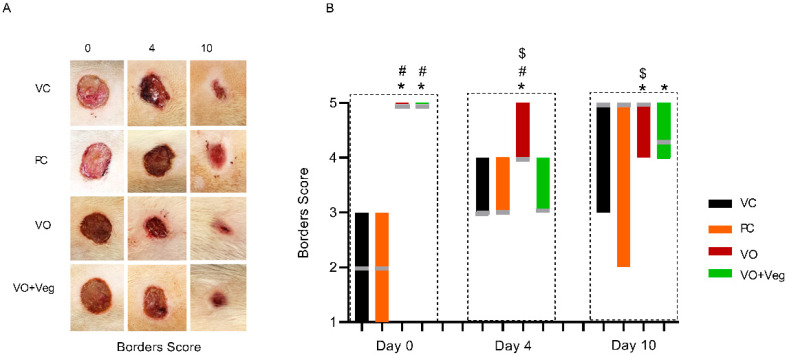
Representative photographs of borders in respective groups and treatment evaluation time (**A**). Graph of dryness scores in VC, PC, VO, and VO + Veg groups, in three periods of evaluation. The bars represent the data dispersion (minimum and maximum), and the horizontal gray lines represent mode (**B**). * *p* < 0.05: VO vs. VC; # *p* < 0.05: VO vs. PC, and $ *p* < 0.05: VO vs. VO + Veg.

**Figure 6 antioxidants-11-01647-f006:**
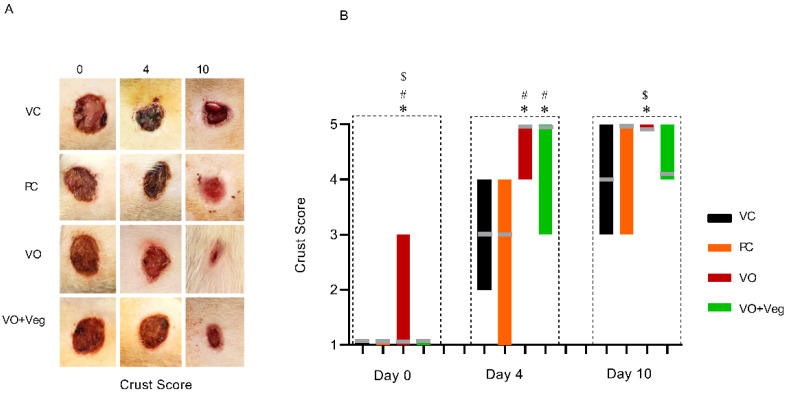
Representative images of crust in respective groups and treatment evaluation time (**A**). Graph of crust scores in VC, PC, VO, and VO + Veg groups, in three periods of evaluation. The bars represent the data dispersion (minimum and maximum), and the horizontal gray lines represent mode (**B**). * *p* < 0.05: VO vs. VC; # *p* < 0.05: VO vs. PC, and $ *p* < 0.05: VO vs. VO + Veg.

**Figure 7 antioxidants-11-01647-f007:**
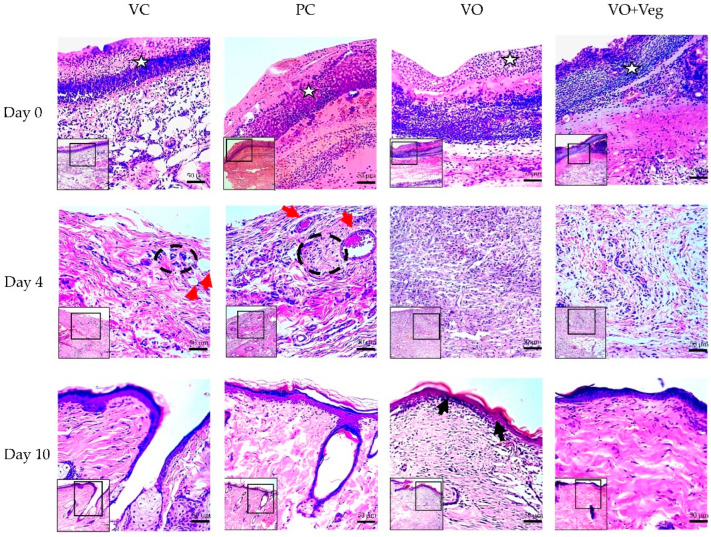
Histological microscopic representative images of the wound healing process of the experimental groups VC, PC, VO, and VO + Veg. Skin sections demonstrate dermis and epidermis stained with hematoxylin and eosin at 10× (smallest figure) and 20× (larger figure) magnifications. The white stars indicate the granulation tissue region. Red arrows indicate perivascular inflammatory infiltrate in the upper dermis, with dotted black demarcation lines of regions rich in inflammatory cells. On day 10, the epidermis layer (upper edge) is highlighted, with a keratin-rich stratum corneum, mainly in the group, indicated by the black arrows.

**Figure 8 antioxidants-11-01647-f008:**
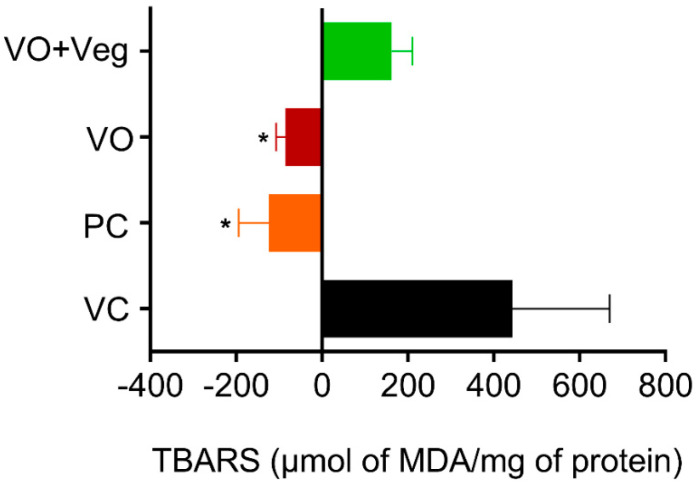
Delta levels of lipid peroxidation (TBARS) in skin experimental groups. Data are expressed as mean ± SEM * *p* < 0.05. One-way ANOVA followed by Tukey’s post hoc test.

**Figure 9 antioxidants-11-01647-f009:**
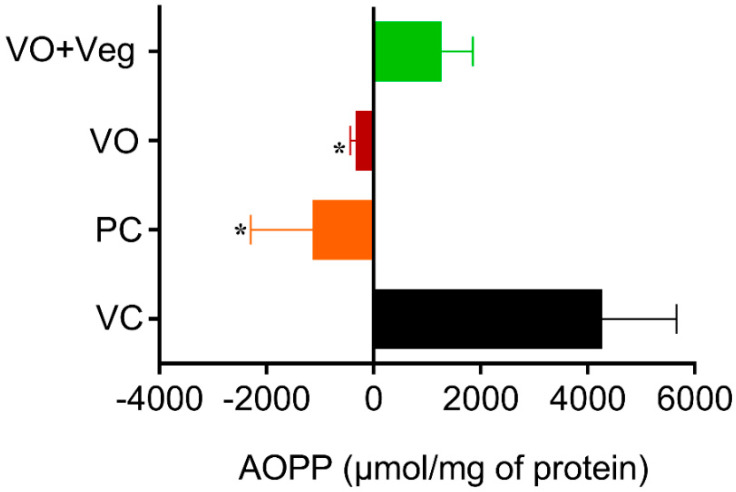
Delta levels of advanced oxidation protein products (AOPP) in skin experimental groups. Data are expressed as mean ± SEM * *p* < 0.05. One-way ANOVA followed by Tukey’s post hoc test.

**Figure 10 antioxidants-11-01647-f010:**
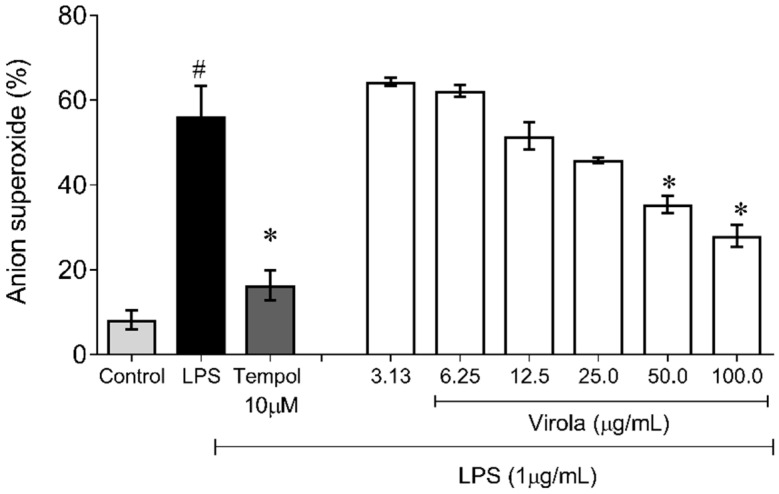
Lipopolysaccharide (LPS)-induced superoxide anion production showing that RV only at high doses (V50 and V100) inhibited superoxide anion induced by LPS in macrophages. Tempol: water soluble superoxide dismutase mimetic (12 mM). The values are presented as the mean ± SEM for *n* = 3 samples in triplicate. # *p* < 0.05 vs. control group. * *p* < 0.05 vs. LPS group (one-way ANOVA).

**Figure 11 antioxidants-11-01647-f011:**
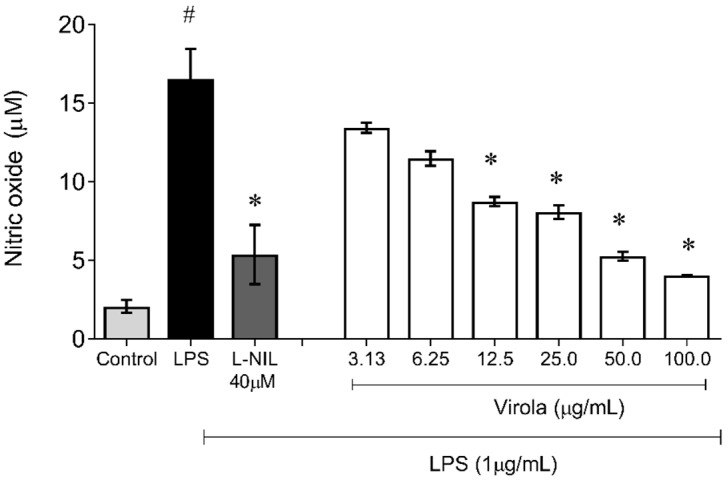
Lipopolysaccharide (LPS)-induced nitric oxide (NO) production showing that RV from medium dose (V12.5) until high dose (V100) inhibited NO production induced by LPS in macrophages. L-NI: selective inhibitor of inducible nitric oxide synthase (50 µM). The values are presented as the mean ± SEM for n = 3 cells per group. # p < 0.05 vs. control group. * *p* < 0.05 vs. LPS group (one-way ANOVA).

**Table 1 antioxidants-11-01647-t001:** Variation in body weight (g) of Wistar rats after treatment with topical VO and other groups for 10 consecutive days.

Body Weight (g)					
Groups	Vehicle (VC)	Positive Control (PC)	Virola 5% (VO)	Virola + Vegederm (VO + Veg)	*p*
Before the injury	388 ± 37	359 ± 48	353 ± 7	336 ± 29	0.925
Day 0	360 ± 33	334 ± 49	317 ± 27	337 ± 27	0.861
Day 4	335 ± 26	342 ± 50	322 ± 23	345 ± 28	0.964
Day 10	375 ± 31	347 ± 40	338 ± 19	353 ± 24	0.878

Data are expressed as mean ± SEM. Data were compared by one-way ANOVA.

**Table 2 antioxidants-11-01647-t002:** Serum biochemical parameters in experimental healing groups of Wistar rats.

Parameters	Vehicle (VC)	Positive Control (PC)	Virola 5% (VO)	Virola + Vegederm (VO + Veg)	*p*
	**Day 0**				
ALT (U/L)	44 ± 6	39 ± 7	28 ± 3	41 ± 3	0.169
AST (U/L)	134 ± 22	97 ± 13	72 ± 13	97 ± 15	0.101
Urea (mg/dL)	51 ± 3	46 ± 1	49 ± 2	40 ± 1	0.436
Urea nitrogen (mg/dL)	24 ± 1	22 ± 1	23 ± 1	24 ± 1	0.552
Serum creatinine (mg/dL)	0.33 ± 0.01	0.32 ± 0.01	0.36 ± 0.01	0.34 ± 0.03	0.500
	**Day 4**				
ALT (U/L)	35 ± 2	33 ± 5	40 ± 5	56 ± 9	0.061
AST (U/L)	120 ± 9	95 ± 12	133 ± 14	161 ± 50	0.406
Urea (mg/dL)	53 ± 3	41 ± 4	51 ± 9	50 ± 2	0.446
Urea nitrogen (mg/dL)	24 ± 2	19 ± 2	24 ± 4	23 ± 1	0.541
Serum creatinine (mg/dL)	0.40 ± 0.01	0.25 ± 0.06	0.32 ± 0.02	0.35 ± 0.01	0.424
	**Day 10**				
ALT (U/L)	75 ± 29	74 ± 15	39 ± 4	40 ± 5	0.251
AST (U/L)	217 ± 73	263 ± 105	120 ± 28	119 ± 34	0.363
Urea (mg/dL)	42 ± 4	54 ± 4	42 ± 6	45 ± 3	0.207
Urea nitrogen (mg/dL)	20 ± 2	25 ± 2	20 ± 3	21 ± 1	0.204
Serum creatinine (mg/dL)	0.35 ± 0.03	0.37 ± 0.02	0.28 ± 0.02	0.24 ± 0.06	0.422

Data are expressed as mean ± SEM. Data were compared by one-way ANOVA.

## Data Availability

The data is contained within the article.

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
