# Peer review of "Development and Evaluation of *Virola oleifera* Formulation for Cutaneous Wound Healing"

_antioxidants, 2022, doi:10.3390/antiox11091647_

Round 1

Reviewer 1 Report

For the Manuscript: antioxidants-1838175 minor revisions are required.

- it would be optimal to know the composition of the VO extract to identify which molecules are most active in the skin damage repair process, the authors stated that this would be done in the future. Still, the molecular identification of the components of the extract remains fundamental.

- the histological investigation would be complete with an in-depth analysis of the investigated samples in electron microscopy.

- how were the concentrations used in the study from 3.13 to 100ug / ml chosen? Was a dose-response curve used?

- The values reported in Tab.2 would be more explicit in the histogram.

-In the legends of fig. 5 and fig. 6 the statistic marked $ is indicated, of which the reference is not indicated.

- There are typos in the text, and the bibliography must be standardized and formatted.

Reviewer 2 Report

The Authors presented the study of the possible use of VO as an active agent for wound healing. The topic, as well as the results, might be presented in the Antioxidants Journal. Nevertheless, major revision is necessary.

First of all, please specify, how did you choose the optimal composition of the cream? If needed add supporting materials, or add them to the text.

The results presented on mice are interesting and worth presenting.

Detailed comments are listed below:

Could you highlight what kind of the major compound/compounds has the biggest impact on wound healing?

Did you or someone else make detailed studies of VO using i.e. some chromatographic or spectrophotometric studies? If yes, please add it/or highlighted it in the manuscript.

Abstract

Add information that you have prepared cream.

Please check if the word moisture/ointment will be more suitable as a name of the prepared composition (you can leave cream as is – this remark is not mandatory)

Line 23 please check compared-control does "-" necessary?

Check words dividing "-" in the whole text

Introduction

This section could be extended. You can add here some examples of plants with proven wound healing properties

Line 36 this sentence is more suitable for the end of the introduction or should be inserted in the results and discussion section

There are several wound kinds. Please add some general information about the current use methods. It will also extend the introduction as I suggested above

Materials and methods

I really adore figure 1.

Line 78 healing effect of what

Please give a table with the moisture composition

Please add some comments on what was the base for the selected composition. Did you make some preliminary tests? Or did you base it on the previous studies, etc.?

line 149 - please check the numbering of the headlines

Results and discussion

What does mean SEM here?

Please consider putting the whole names and short names in the table heading

line 198 is it necessary to use the word "remarkable"? Please use less emphasized words

in line 214 what do you mean here by dryness?

line 354/355 add some examples

Conclusions

Please rewrite the first sentence. You do not have to highlight "for the first time". Add information on the future perspectives, i.e. what about the possible use of different matrices, etc.

Round 2

Reviewer 2 Report

The Authors updated the manuscript due to my comments. The manuscript content is interesting and worth presenting in the Journal. The most important information from the results is the possible development of the natural-based composition for further wound healing applications.

I m strongly recommended this manuscript for publishing.

Here are two editorial remarks, which doesn't affect the manuscript quality:

-check lines 266, 278 "$p". Could you change the sign "$" for some other sign

Figure 7 in my opinion should be enlarged in the final version